

# Multi-model analysis of the Adriatic dense water dynamics

Petra Pranić[1], Cléa Denamiel[2], Ivica Janeković[3], Ivica Vilibić[2]

[1]Institute of Oceanography and Fisheries, Šetalište I. Meštrovića 63, 21000 Split, Croatia
[2]Ruđer Bošković Institute, Division for Marine and Environmental Research, Bijenička cesta 54, 10000 Zagreb, Croatia
[3]Ocean Graduate School and the UWA Oceans Institute, The University of Western Australia, 7 Crawley, WA, Australia

*Correspondence to*: Petra Pranić (pranic@izor.hr)

ORCiD: 0000-0001-6642-033X

**Abstract**

This study aims to enhance our understanding of the bora-driven dense water dynamics in the Adriatic Sea using different state-of-the-art modelling approaches during the 2014-15 period. Practically, we analyse and compare the results of four different simulations: the latest reanalysis product for the Mediterranean Sea, the recently evaluated fine-resolution atmosphere-ocean Adriatic Sea climate model and the long-time running Adriatic Sea atmosphere-ocean forecast model used

in both hindcast and data assimilation (with 4-day cycles) modes. As a first step, we evaluate the resolved physics in each simulation by focusing on the performance of the models. Then, we derive the general conditions in the ocean and the atmosphere during the investigated period. Finally, we analyse in detail the numerical reproduction of the dense water dynamics as seen by the four simulations. This study confirms that kilometre-scale atmosphere-ocean approach, non-hydrostatic atmospheric models, fine vertical resolutions in both atmosphere and ocean and proper location and forcing of the

open boundary conditions are prerequisites for appropriate modelling of the ocean circulation in the Adriatic basin, which then may be improved by a data assimilation method. As proof, the 31-year long evaluation run of the Adriatic Sea climate model which meets these requirements is found to be able to outperform most aspects of the reanalysis product, the short-term hindcast and the data assimilated simulation, in reproducing the dense water dynamics in the Adriatic Sea.

## 1 Introduction

The focus of this study is the Adriatic Sea — an elongated semi-enclosed basin located in the northern Mediterranean Sea. The main geomorphological features of the Adriatic Sea (Fig. 1a) include a shallow bathymetry of the northern Adriatic shelf, gradually increasing in depth towards the 280 m deep Jabuka Pit. The middle Adriatic is separated from the ~1200 m deep Southern Adriatic Pit (SAP) by the Palagruža Sill, whereas in the very south, the Otranto Strait connects the Adriatic with the northern Ionian Sea. The Adriatic region is also characterized by an extremely complex eastern coastline with many islands

and large mountain ranges along the entire basin.



The thermohaline circulation is one of the main factors influencing the Adriatic Sea dynamics. On the one hand, the river Po in the northern Adriatic drives the outward Western Adriatic Current (WAC) along the western coast. On the other hand, the inward Eastern Adriatic Current (EAC) flows along the eastern side of the Adriatic Sea and transports the water masses from the Mediterranean Sea and, in coastal regions, from large rivers located in northern Albania and southern Croatia. Another

important driver of the thermohaline circulation is the formation of the densest water mass in the whole Mediterranean Sea, the North Adriatic Dense Water (NAdDW, Zore-Armanda, 1963). It is known to occur in winter during severe bora events associated with hurricane-strength gusts up to 50 m/s (Belušić and Klaić, 2004). Bora events have a typical duration of about two days and up to a week (Belušić et al., 2004; Grisogono and Belušić, 2009; Stiperski et al., 2012). They are strongly influenced by the orography and occur most frequently and most intensely along the northern Velebit mountain (e.g., Belušić

et al., 2007; Gohm et al., 2008; Grubišić, 2004; Klemp and Durran, 1987; Kuzmić et al., 2015; Smith, 1987). Practically, the alternation of major mountain gaps and peaks along the Velebit mountain range results in the formation of gap jets and wakes (e.g., Alpers et al., 2009; Jiang and Doyle, 2005; Signell et al., 2010). The bora jets are thus occurring in known locations (Fig. 1b). The Trieste Jet has the northernmost location, the Senj Jet is the most intense and furthest reaching at sea, while the Karlobag and Sukošan Jets are strongly impacting the upper middle Adriatic (Dorman et al., 2007; Pullen et al., 2007;

Janeković et al., 2014; Denamiel et al., 2020a, 2020b). Several jets also occur along the eastern coast in the middle and southern Adriatic (Grisogono and Belušić, 2009; Horvath et al., 2009). During the most extreme bora events, the intensity of the upward sea surface heat fluxes – taking out heat from the sea – is largely increased along the jets inducing negative buoyancy fluxes associated with sea surface cooling at hourly to daily time scale (e.g., Janeković et al., 2014; Denamiel et al., 2021a). This cooling, in addition to the homogenization of the coastal waters during the late autumn and winter seasons, results in the

formation of dense waters over the northern Adriatic and the Kvarner Bay (e.g., Janeković et al., 2014; Ličer et al., 2016; Vilibić et al., 2018). Days to weeks after such bora events, a strong thermohaline circulation – mostly driven by bottom density currents – starts in the Adriatic-Ionian basin and generally last for months (Orlić et al., 2007). Indeed, the dense waters, generated in the northern Adriatic or within the Kvarner Bay, travel along the Italian coast following the Po River plume (Artegiani and Salusti, 1987; Vilibić and Mihanović, 2013) and either leave the Adriatic basin towards the northern Ionian Sea

or are collected within the Jabuka Pit (Marini et al., 2006), the SAP (Querin et al., 2016), but also within the Kvarner Bay, which serves as both an area of generation and deposition of the dense waters. Thus, bora-driven dense water formation in the northern Adriatic, jointly with the deep waters generated through bora-driven open convection in the SAP (Gačić et al., 2002), has a crucial impact on the Adriatic thermohaline circulation as well as biogeochemical properties (Boldrin et al., 2009, Bensi et al., 2013; Gačić et al., 2010; Batistić et al., 2014; Jasprica et al., 2022).

Past numerical studies have shown the importance of several factors influencing the modelling of the bora-driven dense water formation. At first, the atmospheric forcing used as a source of forcing for the ocean models were not capable to properly reproduce the extreme bora events driving this process (Bergamasco et al., 1999; Vested et al., 1998; Beg-Paklar et al., 2001; Zavatarelli et al., 2002; Oddo and Guarnieri, 2011). Indeed, the resolution of the atmospheric model has been found to be one





of the most important characteristics known to impact the bora wind speeds, due to an improved reproduction of the orography
and the enhancement of jet flows in finer grids (Belušić et al., 2017). Also, the importance of the ocean model resolution has
been demonstrated through many studies that used kilometre-scale limited-area models to simulate ocean processes driven by
extreme conditions in the Adriatic Sea (e.g., Cavaleri et al., 2010, 2018; Ricchi et al., 2016; Carniel et al., 2016; Denamiel et
al., 2020b). Further, the influence of the freshwater forcing in the ocean models was found to be crucial in modelling the dense
water formation. Specifically, the river climatology used in previous studies overestimated real river discharges along the
eastern Adriatic coast (Janeković et al., 2014) and has been replaced by a new climatology. That significantly improved the
reproduction of the dense water formation, in particular at its secondary source location in the Kvarner Bay (Vilibić et al.,
2016; Mihanović et al., 2018; Vilibić et al., 2018). Other factors such as the choice of open boundary conditions, the
parametrization of vertical mixing and diffusion, etc., were also found to be important (Benetazzo et al., 2014; Janeković et
al., 2014).

The proper representation of the bora-driven dense water dynamics in the Adriatic Sea is still nowadays challenging, whether
it is for research purpose as hindcast simulations and reanalysis products or for operational purpose as forecast simulations.
This is why, recently, data assimilation was explored as an avenue to improve free model simulations, including the dense
water dynamics in the Adriatic Sea (Yaremchuk et al., 2016; Janeković et al., 2020). More particularly, the most advanced
variational scheme, the Four-Dimensional Variational (4D-Var; Courtier et al., 1994; Janeković et al., 2013; Iermano et al.,
2015; Sperrevik et al., 2017; Janeković et al., 2020) was used during the 2014-15 period when a large number of in situ salinity,
temperature and current observations were available (Janeković et al., 2020). Further, a 31-year evaluation simulation (1987-
2017 period) of a climate model using kilometre-scale atmosphere-ocean models over the Adriatic basin has also been recently
completed and evaluated (Pranić et al., 2021; Denamiel et al., 2021b). The aim of this study is thus to compare the currently
available modelling strategies used to represent the bora-driven dense water dynamics in the Adriatic Sea. The approaches
85    considered in the study are the newest reanalysis product for the Mediterranean Sea, the recently evaluated fine-resolution
atmosphere-ocean Adriatic Sea climate model (Pranić et al., 2021; Denamiel et al., 2021b) and the year-long atmosphere-
ocean Adriatic Sea forecast model used in both hindcast mode and with a 4D-Var data assimilation procedure (Janeković et
al., 2020).

In the following section, the numerical models as well as the methods used to perform this study are described. The results of
90    the analyses are presented in Sect. 3 and discussed in detail in Sect. 4. Finally, the main conclusions of the study are summarized
in Sect. 5.





## 2 Material and methods

### 2.1 Northern Adriatic Experiment (NadEx)

The time period investigated in this study includes the Northern Adriatic Experiment (NadEx) campaign which took place
between late autumn 2014 and summer 2015. The aim of the NadEx campaign was to study the dense water generation and
transport within and off the Kvarner Bay. It consisted in collecting temperature, salinity and current data using several
instruments and observing platforms. To measure the currents, acoustic Doppler current profilers (ADCPs) were deployed at
9 locations between late November 2014 and mid-August 2015, while conductivity–temperature–depth (CTD) probes
measured salinity and temperature at 5 of the ADCP locations. Additionally, vertical profiles of temperature and salinity were
acquired at 19 CTD stations during two cruises between 3 and 6 December 2014 and between 26 and 29 May 2015. An ocean
glider was operated off the Kvarner Bay in a campaign lasting only for 3 days – from 24 to 27 February 2015, while an Arvor-
C profiling float was deployed on 19 February 2015 in the northern part of Kvarner Bay and was recovered on 15 March 2015
off the Istria coast. The full description of the NadEx campaign is provided in Vilibić et al. (2018). During the campaign, three
severe bora episodes with gusts above 50 m s⁻¹ in the Velebit channel occurred: between 28 December 2014 and 1 January
2015, between 4 and 7 February 2015 and between the 3 and 6 March 2015. The NadEx campaign was thus able to partially
observe the dense water generation within the Kvarner Bay during the 2014-15 period.

Due to the unique dataset collected during the NadEx campaign – which has already been used either in data assimilation
experiments (Janeković et al., 2020) or in evaluation studies (Vilibić et al. 2018; Pranić et al. 2021) – the 2015 bora events
present a unique opportunity to compare the capacity of various types of models (e.g., reanalysis, hindcast, assimilated
simulations, climate simulations) to reproduce the dense water dynamics in the Adriatic basin.

### 2.2 Numerical models

#### 2.2.1 Mediterranean Sea reanalysis

The newest high-resolution physical reanalysis product for the Mediterranean Sea (hereafter referred as MEDSEA) distributed
within the Copernicus Marine Environment Monitoring Service (CMEMS) framework (Escudier et al., 2020, 2021) is used in
this study. It covers the whole Mediterranean Sea and a part of the Atlantic Ocean for the 1987-2019 period. The reanalysis is
generated by the Nucleus for European Modelling of the Ocean (NEMO) V3.6 model (Madec et al., 2017). NEMO is a non-
linear, free surface, z-coordinate model which solves the primitive equations using time-splitting techniques. The model has a
horizontal resolution of 1/24° (4.5 km) and 141 unevenly distributed vertical levels (thickness varies from 2 m in the upper
layers to 100 m in the deeper layers). The atmospheric forcing of the ocean model is provided by the European Centre for
Middle-range Forecast (ECMWF) ERA5 reanalysis with 0.25° horizontal resolution and 1 h temporal resolution (Hersbach et
al., 2020). The initial conditions are taken from the SeaDataNet (https://www.seadatanet.org/, last access: 14 November 2022)
climatology.





The model is combined with a three-dimensional variational (3D-Var) data assimilation scheme OceanVar (Dobricic and Pinardi, 2008) in which the in-situ data from CTD, Argo profiling floats – ARGO (https://argo.ucsd.edu, last access: 1 May

2022) and expendable bathythermograph (XBT) measurements are assimilated into the model along with satellite altimetry observations. Furthermore, a modified GEBCO 30arc-second grid (Weatherall et al., 2015) is used for topography. Nudging schemes are used to constrain heat and freshwater fluxes towards sea surface temperature (SST) and salinity observations. Also, a large-scale bias correction scheme is added to correct the model tendencies. The runoff of the 39 rivers is obtained from monthly mean climatological datasets. Further, SST fields are used for the correction of surface heat fluxes with a

Gaussian relaxation coefficient dQ/dSST.

Finally, MEDSEA reanalysis, assessed through a comparison against in-situ and satellite observations as well as climatologies, showed a better representation of the main dynamics of the Mediterranean region than the previous, lower-resolution (1/16°) reanalysis (Simoncelli et al., 2016, 2019). A more detailed description of the state-of-the-art MEDSEA product and all the components of the model used to produce it, can be found in Escudier et al. (2021).

2.2.2 ROMS and ALADIN/HR modelling system

ROMS and ALADIN/HR is an atmosphere-ocean modelling system consisting of the ocean model ROMS (Regional Ocean Modelling System; Shchepetkin & McWilliams, 2009) and the atmospheric model ALADIN/HR (Aire Limitée Adaptation dynamique Développement InterNational). It has been operationally integrated since 2008 and has already been evaluated in several studies (e.g., Janeković et al., 2014; Vilibić et al., 2016, 2018).

Regarding the ocean component, ROMS is a 3-D, free surface, bathymetry following, s-coordinate model in which primitive equations are solved with a finite-difference approximation and a time-splitting method (Shchepetkin and McWilliams, 2005, 2009). In this model, the horizontal ROMS grid resolution is 2 km and there are 20 vertically spaced sigma levels controlled by the following parameters: *Vtransform* = 2 and *Vstretching* = 2 with $\theta_s = 7$ (increased resolution at the surface), $\theta_b = 0.5$

, and $h_c = 30$ (critical depth of 30 m). Daily averaged lateral boundary conditions are imposed north of the Otranto Strait from

the Adriatic Forecasting System (AREG, Oddo et al., 2006). AREG is nested within the larger Mediterranean Forecasting System (MFS) which uses 3D-Var data assimilation (Pinardi et al., 2003; Pinardi and Coppini, 2010; Tonani et al., 2014). Furthermore, a bathymetry smoothing achieved with a linear programming (LP) technique (Dutour Sikirić et al., 2009) is used to reduce numerical instabilities. A river discharge climatology is imposed at the freshwater point sources following Vilibić et al. (2016) data.

Concerning the atmospheric model, a hydrostatic version of the ALADIN/HR model is used for the atmospheric forcing at the sea surface. The model is operationally run by the Croatian Meteorological and Hydrological Service (Tudor et al., 2013, 2015) four times a day with initial conditions computed using mesoscale data assimilation. It has a horizontal resolution of 8 km





while the winds are dynamically downscaled to a horizontal resolution of 2 km (Ivatek-Šahdan and Tudor, 2004). In the vertical, 37 sigma layers are used in the model while the temporal resolution of all variables is 3 h. Lateral boundary conditions

are obtained from the operational forecast runs of the Integrated Forecast System (IFS) in the ECMWF, where the global analysis is performed using the 4D-Var data assimilation technique. The transfer of the surface variables into the ocean model is done via bulk parametrization (Fairall et al., 1996).

In the study of Janeković et al. (2020), a year-long (1 October 2014 – 30 September 2015) 4D-Var data assimilation experiment has been applied to the Adriatic Sea. Three model simulations were integrated but only two of them are used in this study: (1)

a non-assimilative, hindcast simulation, hereafter referred as ROMS-hind and (2) a fully assimilative simulation that used all available observations during the 4-days assimilation cycle – hereafter referred as ROMS-full. During the ROMS-full simulation, the physical-space statistical analysis system (PSAS) approach (Moore et al., 2011a) was applied, splitting the one-year simulation into 91 4-day assimilation cycles, each restarting from the previous cycle using saved initial conditions. This window cycling was necessary to ensure the validity of the Tangent Linear model assumption (Powell et al., 2008) within the

4-day window. Concerning the Adjoint model, clamped boundary conditions were used instead of radiation-nudging of 3D fields. High-resolution multi-platform observations were assimilated into the model, including SST measured by satellites, in situ temperature and salinity data measured by various moving (Argo floats, shipborne CTDs, sea glider, towed CTD profiler) and moored platforms, ocean current profiles measured by moored Acoustic Doppler Current Profilers (ADCPs), and 30-minute de-tided surface currents from high-frequency (HF) radars. More information about the observations, experiments and

skill assessment of the model can be found in the study of Janeković et al. (2020).

### 2.2.3 Adriatic Sea and Coast (AdriSC) climate model

The Adriatic Sea and Coast modelling suite (AdriSC; Denamiel et al., 2019) is based on a modified version of the Coupled Ocean-Atmosphere-Wave-Sediment-Transport (COAWST V3.3) modelling system (Warner et al., 2010). The AdriSC climate module has been developed as one of three modules of the AdriSC model, to study long-term kilometre-scale processes in the

Adriatic region (e.g., Denamiel et al., 2022). The performance of the atmospheric (Denamiel et al., 2021b) and ocean (Pranić et al., 2021; Denamiel et al., 2022) components of the climate module has been thoroughly evaluated.

The ocean component is the ROMS model (hereafter referred as AdriSC-ROMS) with a larger 3 km grid and a nested 1 km grid covering the Adriatic and northern Ionian Sea (Fig. 1a). Regarding the vertical resolution, it uses 35 sigma layers transformed ($Vtransform = 2$) and stretched ($Vstretching = 4$) following Shchepetkin and McWilliams (2009) with increased

resolution at the surface ($\theta_s = 6$) and bottom ($\theta_b = 2$) as well as a thickness of 50 m ($h_c = 50$). The high-resolution bathymetry data is provided for both grids by a Digital Terrain Model (DTM; Denamiel et al., 2018). The bathymetry smoothing (with the minimum depth of 2 m) was done with a LP method (Dutour Sikirić et al., 2009) which minimized the roughness factors (rx0 = 0.2) and kept the DTM bathymetric features while the horizontal pressure gradient errors were



reduced. The initial conditions and boundary forcing of the AdriSC-ROMS 3 km model (sea-surface height, barotropic and baroclinic currents, baroclinic temperature and salinity) are provided daily by the Mediterranean Forecasting System (MFS) MEDSEA v4.1 re-analysis (resolution of 1/16°; Simoncelli et al. 2016, 2019) within the Copernicus Marine Environment Monitoring Service (CMEMS). The river forcing consists of 54 river flows in total (only 49 for the 1 km grid). The river flows are vertically distributed between the first 20 sigma levels and monthly climatologies of the river flows are acquired from various databases and studies. Additionally, a dQ/dSST procedure, described in the study by Denamiel et al. (2019), is used to minimize the corrections of the heat fluxes produced by Weather Research and Forecasting (WRF v3.9.1.1) model (Skamarock et al., 2005), while making sure that no artificial SST trends are generated in the shallow parts of the ROMS grids.

The atmospheric component is the WRF model (hereafter referred as AdriSC-WRF) with a grid resolution of 15 km and a nested 3 km grid (Fig. 1.a). The model is nonhydrostatic and consists of 58 vertical levels refined in the surface layer (Laprise, 1992). The physics and parameterisations rely on the optimal configuration of Adriatic high-resolution WRF models (Kehler-Poljak et al., 2017). For the climate simulation, initial conditions and boundary forcing of the WRF 15 km grid are provided by the 6-hourly ERA-Interim reanalysis fields at 0.75° resolution (Dee et al., 2011; Balsamo et al., 2015). In addition, the sea surface temperature (SST) from the ROMS grids is not prescribed to the WRF models since the spatial extension of the ocean grids does not fully cover the WRF 15 km domain. Hence, the SST forcing is provided by the Mediterranean Forecasting System (MFS) reanalysis with 1/16° resolution distributed in the CMEMS (Simoncelli et al., 2019). Finally, the data exchange between the WRF grid and the ROMS models is achieved with the Model Coupling Toolkit (MCT v2.6.0; Larson et al., 2005).

## 2.3 Methods

In order to compare different simulations, model results with grid resolution coarser than 1 km are interpolated to the AdriSC-ROMS 1 km grid using the regridding routines based on the Earth System Modeling Framework (ESMF) software (http://earthsystemmodeling.org/, last access: 1 May 2022). For the ocean simulations, MEDSEA, ROMS-hind and ROMS-full results are regridded to 1 km resolution, while for the atmosphere, ERA5, ALADIN/HR-hind, ALADIN/HR-full and AdriSC-WRF results are all regridded to 1 km resolution.

Hereafter, the bottom potential density anomalies (PDAs) are calculated using the function available within the NCAR Command Language (NCL) library (Levitus et al., 1994a, 1994b; Dukowicz, 2000; https://www.ncl.ucar.edu/, last access: 14 November 2022) and the downward turbulent heat fluxes are computed as the sum of the latent and sensible heat fluxes based on the formulas provided in Denamiel et al. (2020a, 2021a). To be noted, heat fluxes from the ALADIN/HR-full results are also modified by the 4D-Var data assimilation process. Further, in this study dense waters are defined for PDAs equal or larger than 29.2 kg·m⁻³ based on previous research dealing with dense waters in the Adriatic (e.g., Janeković et al., 2014, Vilibić et al., 2016).





A comparative evaluation of the simulations for the 2014-15 period is carried out in the form of probability density functions
of the biases (i.e., differences) between the results of the simulations and the in-situ temperature and salinity observations
extracted from the NadEx campaign (Vilibić et al., 2018), the Palagruža Sill long-term monitoring transect (Institute of
Oceanography and Fisheries, Croatia) as well as the database published by Vilibić (2021) and described in Pranić et al. (2021).
The probability density functions are obtained with a kernel-smoothing method and, for each model, the median and the median
absolute deviation (MAD) of the biases are calculated.

Further, to quantify the general conditions in the ocean and atmosphere throughout the whole 2014-15 period, an analysis of
the extremes is performed. For each of the four simulations, the results are presented as spatial distributions of extremes
accompanied with spatial distributions of their associated timing (in days). This includes the spatial distributions of the
maximum wind stresses and the minimum turbulent heat fluxes in the atmosphere, and of the maximum PDAs, minimum
temperatures and maximum salinities in the ocean.

In order to better capture the dense water dynamics, two different temporal analyses of the results are also performed. First,
time series of daily surface wind stresses and downward turbulent heat fluxes in the atmosphere, and of daily bottom PDAs,
temperatures and salinities in the ocean are presented as the spatial average over different subdomains selected in areas where
dense waters are known to be either generated or accumulated. These subdomains are the northern Adriatic and the Kvarner
Bay for both atmosphere and ocean as well as the Jabuka Pit and the deep Adriatic for the ocean only (Fig. 1a). Also, the daily
bottom PDA time series is presented without the seasonal signal which is removed from the series using the least-squares
method. Second, the time evolution of the spatial distributions of the bottom PDAs is presented at selected dates – 1 March, 1
April, 1 May and 1 June 2015 – and for the whole 2014-15 period as a movie.

An additional analysis (only presented and commented in Supplementary Material) quantifies the total daily volume transport
of the outflowing dense waters across four transects (T1-T4; Fig. 1a). The outward transport is calculated as a double integral
of velocities normal to the transect over the area of the vertical plane of the transect.

## 3 Results

### 3.1 Comparative evaluation during the 2014-15 period

A brief comparative evaluation of the four simulations is performed in order to quantify the skills of the ocean models against
18987 CTD measurements (Fig. 2c). The number of observations depending on the depth is: (1) 6331 for the 0-40 m range,
(2) 5909 for the 40-100 m range, (3) 2130 for the 200-500 m range, and (4) 1577 for the 500-1200 m range. The observations
partially cover the northern Adriatic, the Kvarner Bay, the Palagruža Sill and the SAP.





For the temperature biases (Fig. 2a), ROMS-hind distribution has a median of -0.37 associated with a large peak and a MAD of ± 0.33 °C. ROMS-full distribution has a lower peak and a median of -0.29 ± 0.31 °C. MEDSEA distribution has a median of 0.00 ± 0.84 °C with a heavier tail of positive biases up to 4.5 °C. AdriSC-ROMS distribution has the lowest peak and a median of -0.04 ± 0.61 °C. Therefore, the ROMS simulations systematically underestimate the sea temperature but the assimilation reduces the biases. The AdriSC-ROMS and MEDSEA models overall better reproduce the observed temperatures but have largest MAD due to an overestimation of the temperatures in MEDSEA and both an over- and underestimation of the temperatures between -2 and +2 °C in AdriSC-ROMS.

For the salinity biases (Fig. 2b), ROMS-hind distribution has the lowest peak and a median of -0.16 ± 0.12. ROMS-full distribution has a higher peak and a median of -0.09 ± 0.09. MEDSEA distribution has a median of 0.00 ± 0.36, a tail of negative biases down to -2.0 and a heavy tail of positive biases with a secondary peak at approximately 1.0. AdriSC-ROMS distribution has a slightly higher peak than MEDSEA and a median of 0.02 ± 0.16 with very low probabilities for negative biases below -0.2 but a heavy tail up to around 1.0 and a secondary peak around 0.4. Hence, the ROMS-full and hind simulations both underestimate the observed salinities but the assimilation reduces the biases. The AdriSC-ROMS model tends to overestimate the salinity while the MEDSEA results display the largest over- and underestimations of salinities.

### 3.2 Analysis of the extremes

To analyse how the different models capture the extremes during the 2014-15 period, the spatial distributions of daily maximum wind stresses, daily minimum downward turbulent heat fluxes and their associated timing are presented in Figures 3 and 4, while the spatial distributions of daily maximum bottom PDAs and their associated timing are presented in Figure 5. Additionally, the spatial distributions of minimum temperatures and maximum salinities are provided and described in Supplementary Material (Fig. S1 and S2).

3.2.1 Wind stresses and downward turbulent heat fluxes

It should be noted that ERA5, which is forcing MEDSEA reanalysis, produces very small wind stresses over the whole basin (Fig. 3a), barely reaching 0.4 N m$^{-2}$ in the northern Adriatic, while ALADIN/HR-hind and ALADIN/HR-full wind stress results (Fig. 3c, 3e) are extremely similar despite the variational scheme of the assimilation changing the wind stresses (i.e., the differences between the ALADIN/HR-hind and full wind stresses are at least an order of magnitude smaller than their differences with the other atmospheric models). Further, AdriSC-WRF, which is the only kilometre-scale atmospheric model used in this comparison, overall generates the largest extremes for both wind stresses (> 1.5 N m$^{-2}$; Fig. 3g) and downward turbulent heat fluxes (< -1100 W m$^{-2}$; Fig. 4g). However, for the downward turbulent fluxes, ERA5 produces maximum heat losses (i.e., minimum heat flux values) comparable to AdriSC-WRF (Fig. 4a, 4g) while ALADIN/HR-full maximum heat losses are at least twice smaller than in ALADIN/HR-hind (Fig. 4c, 4e). In fact, ALADIN/HR-full has the smallest maximum heat losses of all simulations and shows a patchy spatial distribution with the smallest values over the middle of northern





Adriatic, barely reaching 750 W m$^{-2}$ in February-March. Consequently, both MEDSEA and ALADIN/HR-full are strongly influenced by the assimilation (e.g., sea surface temperature coming from remote sensing products or variational changes of

the heat flux forcing, respectively). Another important point is that the turbulent heat fluxes are highly influenced by the sea surface temperature and the relative humidity, which are in return influenced by the solar radiation. The maximum heat losses are thus more likely to be found in December 2014-January 2015 – due to a difference in air-sea temperatures of about 3-4 °C having a larger contribution in the downward turbulent heat flux calculation than the intensity of the wind stresses (Fairall et al., 1996) – than in early February/March 2015 when the temperature differences are smaller.

In the northern Adriatic, the Trieste Jet is seen by ALADIN/HR-hind/full and AdriSC-WRF models with wind stress maximums reaching 0.8 N m$^{-2}$ and 1.3 N m$^{-2}$, respectively. It is also important to highlight that the Trieste Jet produced by ALADIN/HR-hind/full is further extended offshore than in the AdriSC-WRF simulation. For the downward turbulent fluxes, ERA5 and ALADIN/HR-hind/full give small intensity along the Trieste Jet (above -600 W m$^{-2}$) between January and March 2015, while AdriSC-WRF reaches -850 W m$^{-2}$ in December 2014.

The largest values of maximum wind stresses are found in the Kvarner Bay and along the Senj Jet for all simulations including ERA5. They reach up to 1.3 N m$^{-2}$ for ALADIN/HR-hind/full and more than 1.5 N m$^{-2}$ for AdriSC-WRF over a far wider area than the other models. In this region, for the downward turbulent heat fluxes, the minimum values are reached by ERA5 (i.e., -900 W m$^{-2}$ despite not reproducing the bora jets) and AdriSC-WRF (below -1100 W m$^{-2}$), while ALADIN/HR-hind and full reach -850 W m$^{-2}$ and barely -750 W m$^{-2}$, respectively.

In the middle Adriatic, high wind stresses up to 1.2 N m$^{-2}$ for ALADIN/HR-hind/full and AdriSC-WRF models are produced along the Karlobag/Sukošan Jets. However, AdriSC-WRF extends the Karlobag Jet to the middle of the Adriatic with values (up to 1.5 N m$^{-2}$) several times larger than achieved with the ALADIN/HR-hind/full simulations. It also produces some strong wind stresses up to 1.3 N m$^{-2}$ along the Dalmatian coast where other bora jets are known to be located. In terms of downward turbulent fluxes, the minimum values are in average -900 W m$^{-2}$ for ERA5, -800 W m$^{-2}$ for ALADIN/HR-hind, larger than -

400 W m$^{-2}$ for ALADIN/HR-full, and below -1100 W m$^{-2}$ along the eastern Adriatic coast for AdriSC-WRF.

In the southern Adriatic, maximum wind stresses in ALADIN/HR-hind/full reach up to 0.7 N m$^{-2}$ but are lower along the coastline. In the AdriSC-WRF simulation, the wind stresses remain relatively small in the southern Adriatic (below 0.5 N m$^{-2}$), aside from a small patch of larger values off the southern Montenegrin coast. For the downward turbulent fluxes, the results obtained with ERA5 and AdriSC-WRF are quite similar with strong intensities along the eastern coast (in average -900 W m$^{-2}$

and -1000 W m$^{-2}$, respectively) and values above -700 W m$^{-2}$ offshore.

Overall, for all models, maximums of wind stresses are associated with bora events, while downward turbulent heat fluxes seem to be more influenced by the seasonal variations of the sea surface temperature. Additionally, the AdriSC-WRF model





generates the strongest dynamics with, in average, the highest wind stresses and the maximum heat cooling, while ERA5 has the lowest wind stresses and ALADIN/HR-full the lowest heat cooling.

3.2.2 Potential density anomalies

In the northern Adriatic, all simulations produce the highest maximum PDA values during late winter (February-March 2015; Fig. 5b, d, f, h). They reach up to 29.4 kg m$^{-3}$ on the shelf for MEDSEA, up to 29.6 kg m$^{-3}$ along the coast but below 29.3 kg m$^{-3}$ on the shelf for ROMS-hind, up to 29.8 kg m$^{-3}$ along the coast and in average 29.5 kg m$^{-3}$ on the shelf for ROMS-full, and finally, above 29.8 kg m$^{-3}$ along the coast and in average 29.7 kg m$^{-3}$ for AdriSC-ROMS (Fig. 5a, c, e, g).

In the Kvarner Bay, both MEDSEA and ROMS-hind have extremely low maximum PDAs (below 29.0 kg m$^{-3}$), indicating no dense water formation in this area. In contrast, both ROMS-full and AdriSC-ROMS give high maximum PDAs (up to 29.6 kg m$^{-3}$ and 29.7 kg m$^{-3}$ respectively). However, ROMS-full presents patch-like PDA distributions with maximums occurring partly during winter and partly during spring, while AdriSC-ROMS has more homogeneous values over the whole Kvarner Bay with maximums occurring mostly in the winter but also in September in a few very small areas. Further, off the Kvarner Bay,
ROMS-full produces a large patch of extremely dense waters (> 29.8 kg m$^{-3}$) which does not seem to be smooth and continuous with the previous data assimilation cycle spatial PDA distributions over the rest of the Adriatic domain. In this case, data assimilation is correcting for the initial state of the ocean model at the start of the assimilation cycle, as the most cost-effective mechanism for correcting suboptimal atmospheric (hydrostatic and coarser) forcing and ocean model vertical and horizontal resolution constraints. This patch occurred in February and is located just southwest from the glider data assimilated in the
model, which is the strongest contributor to the data assimilation cost function at that period (Janeković et al., 2020).

In the middle Adriatic, ROMS-hind shows relatively low maximum PDAs (below 29.1 kg m$^{-3}$) but the other models present some interesting spatial variations. In the Jabuka Pit, which is a known dense water reservoir, maximum PDAs reach up to 29.5 kg m$^{-3}$ in MEDSEA during autumn 2014 (i.e., the highest PDA values over the entire basin), up to 29.6 kg m$^{-3}$ in ROMS-full during spring and summer, and only up to 29.4 kg m$^{-3}$ in AdriSC-ROMS during spring. Additionally, in ROMS-full and
AdriSC-ROMS simulations, the PDA maximums are highest in the western part of the middle Adriatic in late winter and spring, resembling the dense water outflow that normally peaks up to 2 months after its generation in the northern Adriatic. However, in ROMS-full, some high values of maximum PDAs (about 29.4 kg m$^{-3}$) are also present along the Dalmatian islands which is not an area known for the formation or the accumulation of dense waters due to the extensive freshwater discharge of the Neretva River.

In the southern Adriatic, within the SAP and mostly during winter, maximum PDAs reach up to 29.4 kg m$^{-3}$ in MEDSEA, only 29.2 kg m$^{-3}$ in ROMS-hind, up to 29.3 kg m$^{-3}$ in ROMS-full and up to 29.4 kg m$^{-3}$ in AdriSC-ROMS. Along the western side of the SAP, where dense waters are known to cascade through canyon systems (Rubino et al., 2012), ROMS-full and AdriSC-ROMS produce some transport of dense waters (> 29.3 kg m$^{-3}$) mostly in late spring for AdriSC-ROMS and in March for



ROMS-full. Additionally, MEDSEA, ROMS-hind and AdriSC-ROMS present relatively low maximum PDAs (< 29.0 kg m⁻³) in the coastal area east from the SAP, a shelf strongly influenced by the Albanian rivers (Artegiani et al., 1997), while ROMS-full has higher values reaching up to 29.2 kg m⁻³.


Overall, in the northern Adriatic and the Kvarner Bay, where the dense waters are generated during strong bora events, MEDSEA and ROMS-hind have lower maximum PDAs (29.4 kg m⁻³ and 29.6 kg m⁻³, respectively) than ROMS-full and AdriSC-ROMS (29.7 kg m⁻³ and 29.8 kg m⁻³, respectively). However, in AdriSC-ROMS extreme dense waters are generated
homogeneously over the entire northern Adriatic, while they appear as patches in ROMS-full due to 4D-Var data assimilation 4-day cycling which update the initial state of the ROMS model. Surprisingly, in the Jabuka Pit–- a known collector of the dense waters–- the PDAs of ROMS-full are higher than in the AdriSC-ROMS simulation, indicating that either AdriSC-ROMS is far too dissipative, or that the impact of assimilation is high in ROMS-full. Finally, in the SAP, maximum bottom PDAs are produced in all simulations generally during late autumn and early winter (December 2014-January 2015), indicating that
northern Adriatic dense waters didn't reach the bottom of the SAP by the end of any simulation.

### 3.3 Dense water dynamics

3.3.1 Subdomain-averaged time series

To better understand how the different models capture the dense water dynamics within the Adriatic basin, the daily results are presented as time series spatially averaged over the known sites of generation and collection of dense waters (Fig. 6-8 and
Supplementary Material Fig. S3 and S4).

In the northern Adriatic, all models present three prominent peaks of wind stresses (Fig. 6a), capturing the three severe bora events that occur during the NAdEx campaign: 28 December 2014 – 1 January 2015, 3-7 February 2015 and 3-6 March 2015. These dominant wind stress events are also associated with peaks of downward turbulent heat fluxes in all models (Fig. 6c). However, the intensities of the ERA5 wind stress peaks (0.15, 0.3 and 0.2 N m⁻²) are half those in ALADIN/HR-full/hind and
AdriSC-WRF which are all similar (peaks at 0.3, 0.6 and 0.5 N m⁻²). Further, the intensity of the downward turbulent heat flux peaks is often lower and more spread or shifted over time in ALADIN/HR-full (peaks at -300, -450 and -300 W m⁻²) than in the other models, due to the variational scheme used in the assimilation. It should be noted that the strongest peaks in downward turbulent heat fluxes are always reached by ERA5 and/or AdriSC-WRF (peaks at -600, -400 and -350 W m⁻²), while ALADIN/HR-hind produced slightly smaller intensities in general (peaks at -500, -350 and -300 W m⁻²). Concerning the
associated bottom PDA time variations (Fig. 7a), it should be first noted that the AdriSC-ROMS PDAs are systematically higher than in the other models by 0.2-0.8 kg m⁻³ due to higher salinity (differences of about 0.3-0.6; Fig. S4a). Second, for all simulations, the maximum values are obtained between February and March 2015, when the dense water generation is found to occur (Vilibić et al., 2018). Further, in February 2015, a large increase of bottom PDAs – probably driven by the assimilation of the Arvor-C, towed CTD and glider data which influenced two 4-day cycles – is seen in ROMS-full, which reaches values





nearly as high as in AdriSC-ROMS. The PDAs without seasonality show that the peaks of density due to the bora-driven dense water formation are reproduced in all models (Fig. 8a). The highest increases in density during these peaks are always reached by ROMS-full (0.4, 0.35 and 0.3 kg m$^{-3}$) and the lowest by MEDSEA (below 0.1 kg m$^{-3}$ for the 3 peaks). However, the MEDSEA and AdriSC-ROMS densities already increased before the first bora event by 0.2 kg m$^{-3}$ which means that in fact the highest peak is reached by AdriSC-ROMS after the first bora event and that MEDSEA densities are close to AdriSC-

ROMS values. The PDAs without seasonality also clearly show, for all models, a decrease in density during spring and summer when the denser waters are transported from the northern Adriatic towards the south.

In the Kvarner Bay, the three bora peaks of wind stresses (Fig. 6c) and the associated downward turbulent heat fluxes (Fig. 6d) are also seen by all models. However, ERA5 computed wind stresses are always extremely low (below 0.2 N m$^{-2}$) while AdriSC-WRF produces stronger wind stresses (peaks at 0.5, 0.6 and 1.5 N m$^{-2}$) than ALADIN/HR-full/hind (peaks at 0.25,

0.4 and 0.9 N m$^{-2}$). The intensity of the downward turbulent heat flux peaks is again always less and more spread or shifted over time in ALADIN/HR-full (peaks at -400, -400 and -300 W m$^{-2}$) than in the other models (peaks as large as -800, -500 and -600 N m$^{-2}$). Also, AdriSC-WRF models produce eight wind stress peaks above 0.25 N m$^{-2}$ between December 2014 and April 2015, while ALADIN/HR-hind/full only surpasses this threshold for the three main bora events. Consequently, the non-hydrostatic kilometre-scale AdriSC-WRF model (at 3 km resolution) is capable to reproduce much higher wind stresses than

the hydrostatic ALADIN/HR model (at 8 km resolution dynamically downscaled to 2 km for the winds only) due to the impact of the highly non-linear orographic processes on the dynamics of the bora-driven flows (e.g., Grubišić, 2004; Kuzmić et al., 2015). Next, the downward turbulent heat fluxes are less intense in ERA5 and ALADIN/HR-hind than in AdriSC-WRF, indicating that the cooling rates are lower which thus should lead to less generation of dense waters. In terms of bottom PDA analysis (Fig. 6b), similarly to the northern Adriatic subdomain, the AdriSC-ROMS model produces the highest values, while

MEDSEA and ROMS-hind generally have the lowest values with differences up to 0.6 kg m$^{-3}$ in February-March 2015. This difference is again mostly driven by salinity, which is the lowest in MEDSEA and again the highest in AdriSC-ROMS (Fig. S4). However, salinity is much higher in ROMS-full than in ROMS-hind starting in December 2014, when near bottom salinity measurements were available continuously in the Kvarner Bay through the NAdEx campaign. Convincingly, these measurements moved the ROMS-full run from ROMS-hind towards the higher measured salinities and closer to the AdriSC-

ROMS results. As for the northern Adriatic subdomain, the PDAs without seasonality show three main peaks linked to bora-driven dense water formation in all the models (Fig. 8b). However, the timing of the ROMS-full peaks as well as their intensity is generally different than for the other models (which all behave quite similarly), particularly after the second and third bora events. This shows the impact of the assimilation of the NAdEx campaign observations within the ROMS-full model.

In the Jabuka Pit (Fig. 7c and 8c), bottom PDAs (with and without seasonality) from the two free model runs (AdriSC-ROMS

and ROMS-hind) increase from February 2015, when newly generated denser waters from the northern Adriatic start to fill the pit, and peak in late April 2015. However, AdriSC-ROMS PDAs are higher than ROMS-hind both in mean values (more



than 29.1 kg m$^{-3}$ vs. less than 29.0 kg m$^{-3}$) but particularly in increase rates (0.2 kg m$^{-3}$ in 2 months vs. less than 0.1 kg m$^{-3}$ in 2 months) during the known arrival time of dense waters in the Jabuka Pit (i.e., between March and June 2015). Interestingly, ROMS-full shows an earlier increase in PDAs during December 2014 and January 2015, up to 29.3 kg m$^{-3}$, similar to the

values obtained in AdriSC-ROMS in late April. This increase is probably driven by the availability of measurements at that time. Later, after a small decrease between February and March 2015, ROMS-full PDAs start to slowly increase until summer. Different than other simulations, MEDSEA starts with high PDA values in autumn (higher by about 0.2-0.3 kg m$^{-3}$ than other simulations), which then decrease by March down to slightly higher values than ROMS-hind and stabilize till September 2015. This shows that no dense water arrival in the Jabuka Pit is seen by MEDSEA during spring 2015.

In the deep Adriatic (Fig. 7d), bottom PDA values are similar in all models with slightly higher values in ROMS-hind/full and lower values in MEDSEA and AdriSC-ROMS. Further, temporal changes in PDAs are higher in ROMS-full and MEDSEA as they assimilate deep observations (e.g., by Argo profilers up to 700-800 m) which were available during the whole 2014-15 period (Kokkini et al., 2020), as can be clearly seen in the PDAs without seasonality (Fig. 8d).

Overall, the analysis of the time series spatially averaged over the subdomains where dense waters are either generated (i.e.,
northern Adriatic and Kvarner Bay) or collected (Jabuka Pit and deep Adriatic) confirms the results obtained for the extreme values. First, the AdriSC climate simulation generates the strongest dynamics of all the models during the bora events with the highest intensities in wind stress, downward turbulent heat flux and bottom PDA (except in the Jabuka Pit and the deep Adriatic). Second, the MEDSEA model, closely followed by the ROMS-hind model, is generating lowest levels of dense waters during the December 2014-March 2015 period. Finally, the assimilation in ROMS-full, despite reducing the intensity
of the downward turbulent fluxes, tends to increase the bottom PDA values in all the subdomains but particularly in the Kvarner Bay and the Jabuka Pit.

3.3.2 Time evolution of the bottom PDA spatial distributions

To better visualize the evolution in time and space of the dense waters, the spatial distributions of the daily bottom PDAs are analysed both at specific dates – i.e., 1 March (Fig. 9), 1 April (Fig. 10), 1 May (Fig. 11) and 1 June 2015 (Fig. 12) – and for
the entire duration of the 2014-15 period as a movie (provided in the Video Supplement). Hereafter, the results are presented chronologically combining both Figures 9-12 and the movie.

Before the first bora event of 28 December 2014, dense waters are mostly present in the deep Adriatic with bottom PDA values ranging from 29.2 kg m$^{-3}$ for ROMS-hind/full to more than 29.3 kg m$^{-3}$ for MEDSEA and AdriSC-ROMS. However, in the Jabuka Pit, MEDSEA shows PDA values up to 29.5 kg m$^{-3}$ in November 2014, slowly decreasing to 29.2 kg m$^{-3}$ before the
first bora event, but also by ROMS-full around the 20 December 2014 with values below 29.25 kg m$^{-3}$.





During the first bora event, in AdriSC-ROMS (and not in other models) dense waters (above 29.4 kg m$^{-3}$) are immediately generated along the coast of the northern Adriatic (i.e., along the Trieste Jet). Then, these dense waters are transported towards the Po River delta and the northern Adriatic shelf. Denser waters (above 29.45 kg m$^{-3}$) are generated and transported in AdriSC-ROMS from the Gulf of Trieste at the end of January, and also in the Kvarner Bay in both AdriSC-ROMS (with values up to 29.3 kg m$^{-3}$) and ROMS-full (with values up to 29.45 kg m$^{-3}$). Further, in ROMS-full, just before the second bora event, patches of extremely dense waters (above 29.4 kg m$^{-3}$ and up to more than 29.5 kg m$^{-3}$) have grown in the northern Adriatic shelf and of the Kvarner Bay. At the same time, in AdriSC-ROMS, the dense waters start to be transported from the northern Adriatic shelf towards the western Adriatic coast along the Po River plume.

Between the second bora event and the 3 March 2015 (i.e., third bora event), more dense waters are generated in the northern Adriatic (along the Trieste Jet and in the shelf) by all the models, with PDA surpassing 29.5 kg m$^{-3}$ in AdriSC-ROMS and ROMS-hind/full and up to 29.4 kg m$^{-3}$ in MEDSEA. However, it should be noted that MEDSEA only sees dense waters in the northern shelf and not along the Trieste Jet. Further, more dense waters (above 29.5 kg m$^{-3}$) are generated within and off the Kvarner Bay and transported along the Po River plume towards the Jabuka Pit and the southern Adriatic in ROMS-full and AdriSC-ROMS. However, due to the availability of assimilated measurements, ROMS-full first generates dense waters off the Kvarner Bay and then within. In contrast, AdriSC-ROMS clearly transports the dense waters generated within the Kvarner Bay towards the west along the bora jets. On the 1 March 2015 (Fig. 9), dense waters are starting to be collected within the Jabuka Pit in both ROMS-full and AdriSC-ROMS, while no dense water has been transported that far south in MEDSEA and ROMS-hind.

Between the third bora event and 1 April 2015, for ROMS-full and AdriSC-ROMS, after an initial increase along the bora jets, dense waters (above 29.5 kg m$^{-3}$) are transported along the western coast from the northern Adriatic and the Kvarner Bay towards the south, and partially collected in the Jabuka Pit. ROMS-hind also shows some dense water transport (with PDAs barely reaching 29.2 kg m$^{-3}$) from the northern Adriatic towards the Jabuka Pit. However, in MEDSEA, the dense waters generated in the northern shelf (up to 29.45 kg m$^{-3}$) seem to slowly dissipate without being transported. On 1 April (Fig. 10), the northern Adriatic dense waters have decreased to PDA values below 29.3 kg m$^{-3}$ in MEDSEA, barely reaching 29.2 kg m$^{-3}$ in ROMS-hind, being mostly below 29.35 kg m$^{-3}$ in AdriSC-ROMS and have even totally disappeared in ROMS-full. For ROMS-full and AdriSC-ROMS, dense waters (up to 29.35 kg m$^{-3}$ and above 29.5 kg m$^{-3}$ respectively) still remain within the Kvarner Bay.

Between the 1 April and the 1 May 2015, in ROMS-full and AdriSC-ROMS, continuous transport towards the south results in more dense waters being collected in the Jabuka Pit from where they start to cascade towards the SAP via the deepest parts of the Palagruža Sill (Rubino et al., 2012). It should be noted that the cascading occurs along a narrower and more western path in AdriSC-ROMS than in ROMS-full. On the 1 May 2015 (Fig. 11), no dense water is present in the MEDSEA and ROMS-





hind models, except within the SAP. Dense waters (PDA above 29.3 kg m$^{-3}$) remain within the Kvarner Bay, the Jabuka Pit and along the western coast in ROMS-full and AdriSC-ROMS.

Between the 1 May and the 1 June 2015, the remaining dense waters are either transported towards the south or, for the most
part, collected within the Kvarner Bay and the Jabuka Pit in both ROMS-full and AdriSC-ROMS. The collection of dense waters within the Kvarner Bay (particularly in AdriSC-ROMS, where PDAs are above 29.45 kg m$^{-3}$ over most of the bay) can be explained by the fact that this area is much deeper than the open northern Adriatic. On 1 June 2015 (Fig. 12) however, the dense waters collected within the Jabuka Pit have much higher PDAs in ROMS-full (above 29.4 kg m$^{-3}$) than in AdriSC-ROMS (below 29.3 kg m$^{-3}$) despite AdriSC-ROMS clearly producing more dense waters during the three bora events. This
can be explained either by AdriSC-ROMS being too dissipative and/or by the strong impact of the assimilation in ROMS-full.

After the 1 June 2015, dense waters remain within the Kvarner Bay till the end of June in ROMS-full and till the end of September for AdriSC-ROMS, and within the Jabuka Pit till the end of September, with PDA values above 29.25 kg m$^{-3}$ in ROMS-full but barely reaching 29.2 kg m$^{-3}$ in AdriSC-ROMS.

Overall, AdriSC-ROMS generates a larger amount of dense waters than the other models because of the strongest atmospheric
forcing, while MEDSEA and ROMS-hind do not properly reproduce the dense water dynamics in the Adriatic basin. However, ROMS-full collects more dense waters in the Jabuka Pit than all the other models. It can be concluded that AdriSC-ROMS is probably too dissipative during the transport of the dense waters from the northern Adriatic and the Kvarner Bay towards the south. Further, in ROMS-full, the patchy distribution of very dense waters during winter and spring can be explained by the assimilation of data in 4-day cycles for which CTD measurements – collected at some given sites and for some specific days
– took a significant role in adjusting the Adriatic dynamical solutions (Janeković et al., 2020). This demonstrates the importance of the coverage and the long-term availability of the assimilated data. A better representation of the dense water dynamics within the Adriatic basin in ROMS-hind can thus be envisioned (and is possible as demonstrated by the results of the AdriSC model) before performing the data assimilation which, for the moment, is incapable to fully compensate the cumulated weaknesses of the ALADIN/HR+ROMS-hind models.

**4 Discussion**

The multi-model analysis performed in this study has demonstrated that reproducing the dense water dynamics within the Adriatic basin is extremely complex as the presented models produced different or even divergent results despite all being thoroughly evaluated in previous studies (Escudier et al., 2021; Janeković et al., 2104; Vilibić et al., 2018; Pranić et al., 2021; Denamiel et al., 2021b, 2022). However, it is important to keep in mind that the presented results belong to different model
categories: ERA5-MEDSEA is a reanalysis product covering the full Mediterranean Sea for the 1987-2019 period, ALADIND/HR and ROMS does not cover the full Adriatic Sea and is used, in this study, either in hindcast mode (hind) or





fully assimilated mode with 4-day cycles (full) for the 2014-15 period, and finally, AdriSC is the evaluation run of a climate model covering the full Adriatic for the 1987-2017 period. This implies that numerical schemes (e.g., discretization, parametrization) and set-up (e.g., physics, resolution, forcing) used in these models as well as the type of simulation performed
(free run vs. assimilated run) strongly influence the quality of the presented results. As this study only compares state-of-the-art models (ECMWF, WRF, ALADIN in the atmosphere and NEMO, ROMS in the ocean), the differences in numerical schemes will not be discussed hereafter because it is difficult to quantify how they impact the dense water dynamics as they vary from model to model. Though, the different model set-ups will be analysed with the aim to better understand their impact on the bora-driven dense water dynamics in the Adriatic basin.

**4.1 Impact of the resolution and the physics on the bora dynamics**

First, the ERA5 reanalysis at 25 km resolution has been demonstrated to be incapable to capture the bora dynamics (Denamiel et al., 2021a). Consequently, in this study, ERA5 wind stresses are two to three times smaller than the AdriSC-WRF and ALADIN/HR results. However, both in the northern Adriatic and in the Kvarner Bay, heat losses calculated from the ERA5-MEDSEA model – via bulk formulae using sea surface temperature assimilating remote sensing products – are comparable to
the ALADIN/HR-ROMS-hind model (Fig. 5). These heat losses are still underestimated compared to the AdriSC model, particularly within the Kvarner Bay and the Gulf of Trieste as well as along all the bora jets (Fig. 2).

Second, the hydrostatic ALADIN/HR model at 8 km resolution – with the wind fields further dynamically downscaled to 2 km – has already been demonstrated to reproduce the basic bora dynamics (Horvath et al., 2009). However, in the Kvarner Bay region, our results show that the ALADIN/HR wind stresses are not as intense and not covering as wide an area as the
non-hydrostatic AdriSC-WRF model. Indeed, the bora cross-flow variability in the Kvarner Bay might occur at a kilometre scale, in particular during deep bora events (Kuzmić et al., 2015), while bora pulsations have a strong sub-kilometre spatial component, posing a challenge for proper quantification in any kilometre-scale atmospheric model. Nevertheless, Denamiel et al. (2021a) have demonstrated that, during 22 bora events including two in 2015, the AdriSC-WRF 3 km model reproduced very well the wind speed observations at Pula, Rijeka, Ogulin, Zavižan, Gospić and Knin stations (all located in the Kvarner
Bay region) above 20 m/s despite over predicting them by up to 5 m/s below this threshold. Further, the ALADIN/HR-ROMS-hind heat losses are always smaller than those computed from ERA5-MEDSEA and AdriSC models. It is documented that hydrostatic atmospheric models are not capable to capture all the details of the bora jets (Klemp and Durran, 1987; Blockley and Lyons, 1994; Grisogono and Belušić, 2009). Consequently, the hydrostatic approximation used in ALADIN/HR constrains its ability to reproduce the finer-scale details of the bora flow (Horvath et al., 2009). Therefore, heat losses in ALADIN/HR-
ROMS (hind and full) mostly occur along the Senj Jet but are still weaker than in AdriSC (Fig. 2). Further, quite surprisingly, the 4D-Var scheme used in the ROMS-full assimilation is reducing the intensity of the turbulent heat fluxes and thus creates a dynamical imbalance between the wind stresses (which are similar in comparison to the differences between the different atmospheric models) and the heat losses forcing the ocean model.





Finally, the evaluations of AdriSC-WRF model performed both for the climate run over a 31-year period (Denamiel et al.,
2021b) and during extreme bora events (Denamiel et al., 2020a, 2020b, 2021a) have demonstrated that a 3 km resolution is
appropriate to represent the atmospheric dynamics within the Adriatic basin. Further, the results of the AdriSC-WRF model at
3 km resolution (particularly the intensity of the winds) have been shown to converge towards the results obtained with the
higher-resolution AdriSC-WRF-1.5 km model during bora events (Denamiel et al., 2021a). However, only sub-kilometre-scale
atmospheric models can properly capture the highly non-linear dynamics of the bora flows (Kuzmić et al., 2015) and thus
using a 3 km non-hydrostatic model is still a compromise between accuracy and efficiency. This compromise is particularly
important when running multi-year/climate simulations having a tremendous computational cost. This is also highlighted by
Vodopivec et al. (2022), who conducted a sensitivity study over a 16-year period using different runoff configurations and
different sources of atmospheric forcing and concluded that the atmospheric forcing has a substantial impact on the hydrology
and circulation of the Adriatic Sea.

**4.2 Impact of the resolution and the bathymetry on the capacity of the reservoirs to collect dense waters**

In the ocean models, the resolution is mostly going to impact the representation of the many islands located along the eastern
Adriatic coast but more importantly, of the reservoirs collecting the dense waters within the Adriatic basin (i.e., Kvarner Bay,
Jabuka Pit and SAP). Further, different Digital Terrain Models (DTMs) have been used to generate the bathymetries of the
presented models. In order to evaluate the joint impact of resolution and bathymetry, MEDSEA and ROMS-hind/full
bathymetries are compared to the AdriSC-ROMS model at 1 km resolution (Fig. 1b, c). The MEDSEA model is clearly
shallower than AdriSC-ROMS within the Kvarner Bay and the Jabuka Pit (by 60-80 m) but also in the middle of the SAP (by
more than 100 m). Consequently, the capacity of the MEDSEA model to naturally collect the dense waters within the known
Adriatic reservoirs is decreased compared to the AdriSC-ROMS model and thus relies heavily on the assimilation of the
available data. In the ROMS-hind/full model, the bathymetry is also generally shallower than in AdriSC-ROMS within the
Kvarner Bay and along the canyon system between the Jabuka Pit and the SAP (between 20-40 m). This is particularly
important as it might explain the differences in paths seen between ROMS-full and AdriSC-ROMS when the dense waters are
transported from the Jabuka Pit towards the SAP. However, concerning the Jabuka Pit and the SAP, the alternated positive
and negative differences in bathymetry between ROMS-full/hind and AdriSC-ROMS clearly show some shifts in locations.
Whether and how these shifts in location impact the dense water dynamics is not clear with the results presented in this study.
Further, it is important to highlight that the AdriSC-ROMS model uses 35 vertical sigma layers while the ROMS-full/hind
model only has 20 of them. As the bora-driven dense water dynamics requires to properly resolve both the surface (for the sea
temperature cooling) and the bottom (for the dense water transport) layers, the finer vertical resolution used in AdriSC-ROMS
may play a major a role in the overall performance of the model.





**4.3 Impact of the salinity forcing on the dense water generation**

Dense water generation is highly sensitive to the background salinity content provided either through the open boundaries or the direct river outflows imposed on the ocean models.

First, in ROMS-hind, Janeković et al. (2014) quantified an underestimation of salinity by 0.2-0.5 for a simulation of the massive dense water formation in 2012. After updating the old river climatologies that largely overestimated the discharges, Vilibić et al. (2016) confirmed that even the simulations using the most realistic river representation underestimate the background
salinity content within the Adriatic basin. As the AREG model (forcing ROMS-full) is set-up with the old river climatologies and has a low salinity content over the entire Adriatic basin, far too much fresh water is inputted through the ROMS-hind open lateral boundary located in the southern Adriatic. Consequently, the ROMS-hind results presented in this study for the 2014-15 period have low basin-wide salinities and therefore generate dense waters with lower bottom PDA values.

Next, the AdriSC-ROMS model has been thoroughly evaluated over a 31-year period in Pranić et al. (2021). First, in the
northern Adriatic, despite a lack of accuracy for salinities under 36, due to the Po River misrepresentation, the AdriSC-ROMS model has been shown to perform well in reproducing dense water masses. Second, in the Kvarner Bay, AdriSC-ROMS salinities have been demonstrated to be too high, which could lead to a general overestimation of the dense water bottom PDAs in this region. And finally, in the SAP, the evaluation revealed that the salinities and the densest waters are captured relatively well by the AdriSC-ROMS model.

Finally, salinities in MEDSEA are closer to the AdriSC-ROMS results in the southern Adriatic (i.e., Jabuka Pit and deep Adriatic subdomains) and to the ROMS-hind results in the northern Adriatic (i.e., northern Adriatic and Kvarner Bay subdomains) during the entire 2014-15 period (Fig. S4). It can thus be safely assumed that the old river climatologies used in MEDSEA are resulting in low salinities over the northern part of the Adriatic basin and hence lower bottom PDAs during the bora-driven dense water generation events.

**4.4 Impact of the assimilation on the ocean dynamics**

First, in ROMS-full, the 4DVar data assimilation is applied in 4-day cycles which means that the ocean dynamical properties are not continuously smooth in time between the cycles as the ROMS-full model adjusts the initial state at the beginning of each cycle. Consequently, despite the large improvement of the ocean fields used to minimize the cost function of the assimilation, the dense water generation and transport as a continuous process in time is not properly reproduced in ROMS-
full. For example, as the salinity is generally underestimated in ROMS-hind, the data assimilation performed in ROMS-full is constantly trying to adjust salinities (and therefore bottom PDAs) to higher values. However, the data availability is highly variable during the investigated period and, for example, is more concentrated in the Kvarner Bay during the February-March 2015 period or along a northern Adriatic transect (Po-Rovinj) surveyed with a monthly or bimonthly frequency. This thus leads





to having extremely high bottom PDAs present off the Kvarner Bay before the actual generation of the dense waters within
the Kvarner Bay or along the Trieste Jet in the ROMS-full model.

Second, MEDSEA, contrarily to ROMS-full, uses a 3D-Var assimilation approach which is known to lose the temporal information contain in the observations through averaging (Janeković et al, 2020). In general, during the 2014/15 period, MEDSEA assimilates less data than ROMS-full which benefited from the observations collected during the NAdEx campaign. Consequently, MEDSEA is incapable to adjust its solution in order to capture the proper dense water dynamics. For example, in the Jabuka Pit, MEDSEA provides a constant decrease in bottom  PDAs from autumn 2014 to winter 2015 opposite to all the other models and probably driven by the availability of the assimilated observations (e.g., Argo data). However, ROMS-full is likely to have assimilated the same observations within the Jabuka Pit but has also been assimilating Arvor-C and drifter data obtained off the Kvarner Bay during the NAdEx campaign. Further, during the winter, when bora episodes occur, only a small number of SST cloud free scenes are available for assimilation in ERA5. As a result, MEDSEA, contrarily to ROMS-full, is mostly incapable to generate the bora-driven dense waters and hence to transport and collect them within the Jabuka Pit.

## 5 Conclusions

The aim of this study was to enhance our understanding of the bora-driven dense water dynamics in the Adriatic Sea using and analysing different state-of-the-art modelling approaches. The main findings of the study can be summarized as follows:

- In the northern Adriatic and Kvarner Bay, dense water generation is better captured in ROMS-full and AdriSC-ROMS than in MEDSEA and ROMS-hind which are producing lower volumes of dense waters. The AdriSC model generates the strongest dynamics of all the models during the bora events with the largest intensities in wind stresses, downward turbulent heat fluxes and bottom PDAs. Also, extreme dense waters are generated continuously in time and over the entire northern Adriatic in AdriSC-ROMS, while they appear as patches in ROMS-full in which a maximum is found off the southern tip of Istria, along the Senj Jet. This is linked to a combination of parameters including the 4-day cycles of the 4D-Var method used in ROMS-full and the use of atmosphere-ocean kilometre-scale models in AdriSC. Further, in the AdriSC simulation, due to the higher spatial resolution, the densest waters are collected within the Kvarner Bay where they stay for the longest amount of time.
- The transport of dense waters along the western coast is not quantitatively captured by MEDSEA and ROMS-hind. Whereas, in the Jabuka Pit, ROMS-full collects more dense waters than all the other models, indicating that AdriSC-ROMS is probably far too dissipative. Lastly, in the SAP, the results show that the northern Adriatic dense waters did not reach the bottom of the SAP by the end of any simulation, classifying the winter of 2015 as a moderate in dense water formation over the northern Adriatic shelf.





- Impact of resolution of the atmospheric models is best seen in the ERA5 results which strongly underestimate the wind stresses. However, the heat losses are comparable between the models, but generally underestimated compared to AdriSC-WRF. Concerning the hydrostatic approximation, the non-hydrostatic model AdriSC-WRF reproduces more intense wind stresses with larger spatial coverage and stronger heat losses than the hydrostatic ALADIN/HR model.

- The differences in resolution of the ocean models and bathymetry clearly influence the path and deposition of dense waters, yet it is not clear how this impacts the dense water dynamics. Further, the ocean models are highly sensitive to the salinity input which plays an important role in the dense water generation. In particular, the usage of old river climatologies causes lower salinities in ROMS-hind and MEDSEA, hence lower bottom PDAs, while AdriSC-ROMS reproduces higher salinities and PDAs.

- Compared to ROMS-hind, the data assimilation in ROMS-full tends to increase the bottom PDA values in all the subdomains but particularly in the Kvarner Bay and the Jabuka Pit. Although assimilation made a large improvement of the ocean fields, the fields are reflecting initial state adjustments at the beginning of each assimilation cycle hence not producing long temporal smooth transitions. In addition, the lack of vertical resolution in the ROMS-full model probably contributes to the improper representation of the dense water dynamics.

In summary, the reproduction of the dense water dynamics in the Adriatic Sea requires the use of (1) kilometre-scale atmosphere-ocean approach, and non-hydrostatic atmospheric models, (2) fine vertical resolutions in both atmosphere and ocean, (3) proper forcing of the open boundaries of the models, and, finally, (4) appropriate representation of the air-sea interactions (e.g., formulation of the surface wind drag). This study reveals that, if these conditions are fulfilled, models running at the long temporal scales can outperform coarse resolution reanalysis products and assimilated simulations. Nevertheless, in addition to these prerequisites, 4D-Var data assimilation could be used to solve other model problems – such as sea-surface temperature drifts, high mixing of the dense waters, etc. – often found in long-term hindcasts and short-term forecasts. However, such approach would be extremely expensive in terms of the required numerical and observational resources needed to achieve it. This study thus paved the way to a new generation of Adriatic circulation models which now should optimize the accuracy of the results and the usage of the numerical resources.

**Code availability**

The code of the COAWST model as well as the ecFlow pre-processing scripts and the input data needed to re-run the AdriSC climate model in evaluation mode can be obtained under the Open Science Framework (OSF) data repository (Denamiel, 2021) under the MIT license.

**Data availability**

A major part of the observational data set used in this study can be obtained under the Zenodo data repository (Vilibić, 2021)
under the Creative Commons by Attribution 4.0 International license. The remaining part of the observational data set is not publicly available as the data were collected within projects in which they were not publicly disseminated. The data were given for research purposes by the Institute of Oceanography and Fisheries (Croatia) upon request.

The model results used in this study can be obtained under the OSF data repository (Pranić, 2022) under the Creative Commons by Attribution 4.0 International license.

**Video supplement**

The movie of the daily spatial distribution of bottom PDA for the 2014-15 period can be obtained under the OSF data repository (Pranić, 2022) under the Creative Commons by Attribution 4.0 International license.

**Author contribution**

Petra Pranić: Data curation, Formal analysis, Investigation, Visualization, Writing – original draft. Clea Denamiel:
Conceptualization, Data curation, Investigation, Methodology, Resources, Software, Supervision, Writing – review & editing. Ivica Janeković: Methodology, Resources, Software, Writing – review & editing. Ivica Vilibić: Conceptualization, Investigation, Methodology, Supervision, Writing – review & editing.

**Competing interests**

The authors declare that they have no conflict of interest.

**Acknowledgments**

The computing and archive facilities used in this research were provided by the European Centre for Middle-range Weather Forecast (ECMWF) and Pawsey Supercomputer Centre, Australia. The contributions of Copernicus and the Euro-Mediterranean Center on Climate Change are acknowledged. A part of the observational data set used in this research was provided by the Institute of Oceanography and Fisheries (Croatia). The research has been supported by the Croatian Science
Foundation project ADIOS (Grant IP-06-2016-1955) and ADAM-ADRIA (IP-2013-11-5928).

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





**Figure 1.** (a) Models domain with geographical locations, transects and subdomains, (b) AdriSC-ROMS 1 km depths, (c) difference between AdriSC-ROMS 1 km and MEDSEA 1 km bathymetry, (d) difference between AdriSC-ROMS 1 km and ROMS 1 km bathymetry.





**Figure 2:** Probability density functions of the biases between the results of the four simulations and in situ (a) temperature and (b) salinity observations as well as a map with the locations of CTD observations (black dots).





**Figure 3:** Spatial distribution of maximum surface wind stresses and their corresponding timing for (a, b) ERA5, (c, d) ALADIN/HR-hind, (e, f) ALADIN/HR-full and (g, h) AdriSC-WRF.






**Figure 4:** Spatial distribution of maximum downward turbulent heat fluxes and their corresponding timing for (a, b) ERA5, (c, d) ALADIN/HR-hind, (e, f) ALADIN/HR-full and (g, h) AdriSC-WRF.





**Figure 5.** Spatial distribution of maximum bottom PDAs and their corresponding timing for (a, b) MEDSEA, (c, d) ROMS-hind, (e, f) ROMS-full and (g, h) AdriSC-ROMS.





**Figure 6:** Time series of daily turbulent fluxes and wind stresses averaged over two subdomains: (a, c) northern Adriatic and (b, d) Kvarner Bay for the 2014-2015 period and four simulations.




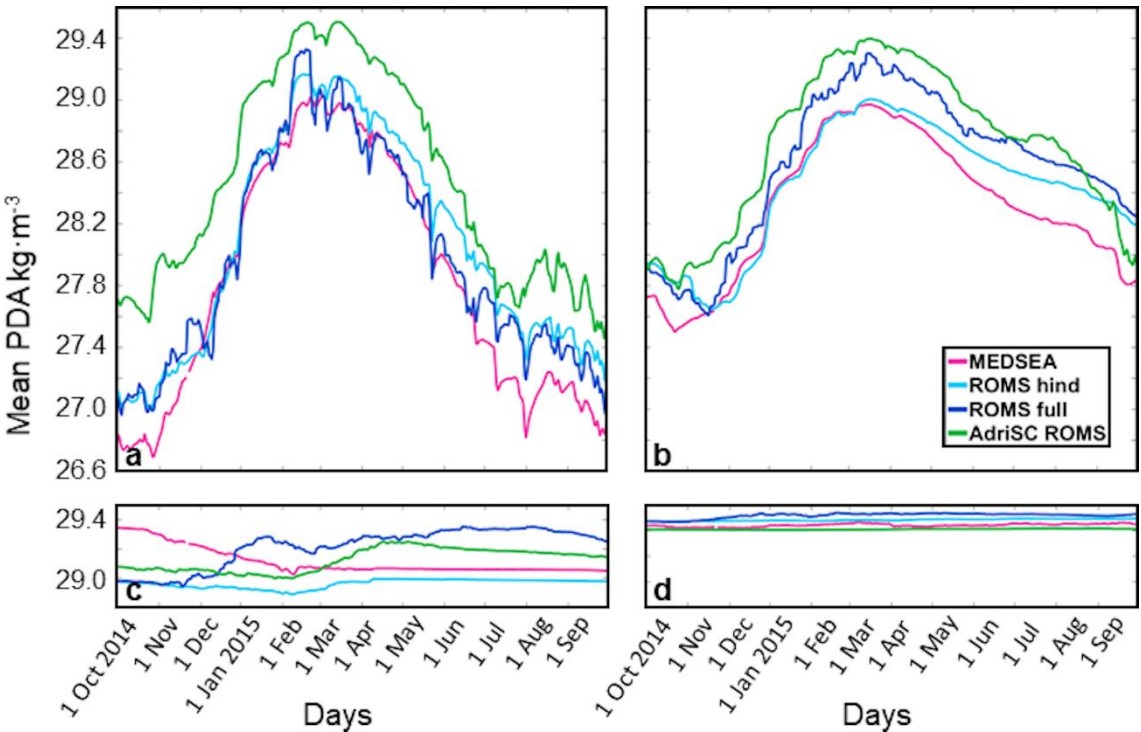

**Figure 7:** Time series of daily bottom PDAs averaged over four subdomains: (a) northern Adriatic, (b) Kvarner Bay, (c) Jabuka Pit and (d) deep Adriatic for the 2014-2015 period and four simulations.





**Figure 8:** Time series of daily bottom PDAs without seasonality averaged over four subdomains: (a) northern Adriatic, (b) Kvarner Bay, (c) Jabuka Pit and (d) deep Adriatic for the 2014-2015 period and four simulations.



**Figure 9:** Spatial distribution of bottom PDAs on 1 March 2015 for (a) MEDSEA, (b) ROMS-hind, (c) ROMS-full and (d)
AdriSC-ROMS simulations.



**Figure 10:** Spatial distribution of bottom PDAs on 1 April 2015 for (a) MEDSEA, (b) ROMS-hind, (c) ROMS-full and (d) AdriSC-ROMS simulations.





**Figure 11:** Spatial distribution of bottom PDAs on 1 May 2015 for (a) MEDSEA, (b) ROMS-hind, (c) ROMS-full and (d) AdriSC-ROMS simulations.





**Figure 12:** Spatial distribution of bottom PDAs on 1 June 2015 for (a) MEDSEA, (b) ROMS-hind, (c) ROMS-full and (d) AdriSC-ROMS simulations.