# Peer review of "Multi-model analysis of the Adriatic dense water dynamics"

_EGUsphere, 2022_

## Author Response (AR1)

Dear Referee #1,

thank you very much for your detailed review and helpful comments which have been addressed in the sections below.

*The transport of dense water is the key element of dense water cascades. In the abstract, the authors claim 'Finally, we analyse in detail the numerical reproduction of the dense water dynamics as seen by the four simulations.' However, the dense water transport is described only qualitatively in the main text, and the assessment of the transport only appears in the appendix as 'additional information'. The transport should be given much greater focus.*

**Response:** The assessment of the transports is added to the main article as subsection 3.3.3. Daily volume transports at transects.

*Lines 18-20 'This study confirms that … are prerequisites for appropriate modelling of the ocean circulation in the Adriatic basin' . This is an overstatement. The study only reveals that one of the four models represents better some variables while other models are better in some other aspects. The paper does not proof that the named parameters are prerequisites for future research. Please re-word.*

**Response:** The sentences have been modified to: „The prerequisites for proper modelling of the ocean circulation in the Adriatic basin, including kilometre-scale atmosphere-ocean approach, non-hydrostatic atmospheric models, fine vertical resolutions in both atmosphere and ocean and the location and forcing of the open boundary conditions, are thus discussed in the context of the analysed simulations".

*Introduction. The authors concentrate entirely on the Adriatic Sea. In order to make the results helpful for a wider oceanographic community the authors are advised to place the bora-driven cascading in a broader context. For example to compare briefly with cascades from other shelves, e.g. as discussed in (Ivanov et al, 2003, https://doi.org/10.1016/j.pocean.2003.12.002; Garcia Quintana et al, 2021. https://doi.org/10.1029/2020JC016951)*

**Response:** The following sentence was added to the Introduction: „Besides the Adriatic Sea, dense water formation on shelfs and its subsequent sinking along shelf breaks (i.e., cascading; Shapiro and Hill 1997, 2003) has been observed and studied in many other areas of the world ocean and particularly in the higher latitudes (Borenas et al. 2002; Shapiro et al., 2003; Wahlin 2002, 2004; Ivanov et al., 2004; Heggelund et al. 2004; Leredde et al., 2007; Garcia-Quintana et al., 2021)".

*Lines 69-70. ' ..the river climatology used in previous studies … has been replaced by a new climatology'. Please clarify, what is the difference between 'old' and 'new' and give a reference. It seems from the text in Line 71 that the 'new' climatology was used in some 'previous studies'.*

**Response:** The sentences have been modified as: „In particular, the river runoff climatology used in previous studies (Raicich, 1994) overestimated real river discharges along the eastern Adriatic coast (Zavatarelli and Pinardi, 2002; Chiggiato and Oddo, 2008) and has been replaced by a new climatology which was based on up-to-date observations (Janeković et al., 2014)".

*Lines 78-79. 'the most advanced variational scheme, the Four-Dimensional Variational…'. Which of the many versions and sub-versions of DA schemes is 'the most advanced' is a matter of discussion. Please re-word.*

**Response:** The sentence was changed to: „More particularly, the Four-Dimensional Variational scheme (4D-Var; Courtier et al., 1994; Janeković et al., 2013; Iermano et al., 2015; Sperrevik et al., 2017) was used during the 2014-15 period when a large number of in situ salinity, temperature and current observations were available (Janeković et al., 2020)".

*Line 81. 'a 31-year evaluation simulation' . Please define what is 'evaluation simulation'. Was it run with or without DA?*

**Response:** The explanation was added to the sentence: „Further, the 31-year (1987-2017) evaluation simulation of the Adriatic Sea and Coast (AdriSC; Denamiel et al., 2019) climate model using kilometre-scale atmosphere-ocean models over the Adriatic basin has also been recently completed and evaluated (Pranić et al., 2021; Denamiel et al., 2021b). This kind of simulations, also referred as "Control Run" in the climate community, produce several decade long simulations forced by reanalysis products (without data assimilation) and are mainly used for evaluation purpose in climate studies. As a free run (i.e., dynamically consistent over decades contrarily to reanalysis products which depend on the availability of the observations; Thorne and Vose, 2010), the AdriSC evaluation simulation has already provided invaluable information about the, till now unknown, kilometre-scale present trends and variability of the Adriatic Sea (Tojčić et al., 2023)."

*Line 85 'the newest reanalysis product for the Mediterranean Sea' . Please give a reference.*

**Response:** References Escudier et al. (2020, 2021) were added.

*Line 95 and Line 98 '…between late autumn 2014 and summer 2015' . '…between late November 2014 and mid-August 2015' . Please exact dates as you do in Line 100.*

**Response:** Regarding the dates, not all ADCPs were moored and recovered at the same cruises and dates (as many institutions were involved in the campaign), so the Authors prefer to keep the formulation as it is, as more details will unnecesarily weighten the article.

*Line 109. '…of various types of models'. This statement is too wide. Based on the results presented in the MS, the authors could only assess, contrast and compare the specific models they used, not the 'types of models'.*

**Response:** The statement is changed to: „…of different models…".

*Line 113.' reanalysis product for the Mediterranean Sea'. Please give the exact product ID. Is it available from CMEMS catalogue? Is it MEDSEA_MULTIYEAR_PHY_006_004 ? If yes, please use the name for product given by the originators, namely 'Med MFC' to avoid confusion. This product is generated not just by NEMO as stated in Lines 115-16 but 'The Med MFC physical reanalysis product is generated by a numerical system composed of an hydrodynamic model, supplied by the Nucleous for European Modelling of the Ocean (NEMO) and a variational data assimilation scheme (OceanVAR)' (https://data.marine.cop ernicus.eu/product/MEDSEA_MULTIYEAR_PHY_006_004/description)*

**Response:** Yes, the reanalysis product is MEDSEA_MULTIYEAR_PHY_006_004. The name of the product stated in Escudier et al. (2020) is „Med MFC". The Authors agree that „Med MFC" is the best name to avoid confusion. However, Escudier et al. (2021) referred to the same reanalysis as „MEDREA24" for the purpose of their study. Thus, the Authors chose the name „MEDSEA" just for the purpose of this study and would like to keep it for practical reasons. The product descripton is added to the Introduction and changed to: „(1) the newest high-resolution physical

reanalysis product for the Mediterranean Sea (Escudier et al., 2020, 2021), hereafter referred as MEDSEA, which is generated by a numerical system composed of the Nucleus for European Modelling of the Ocean (NEMO) V3.6 model (Madec et al., 2017) and a variational data assimilation scheme OceanVAR (Dobricic and Pinardi, 2008), and forced by the ERA5 reanalysis (Hersbach et al., 2020)".

*Lines 113-200. The description of models from previous studies is too extensive, it should be reduced in size and moved from Material and methods to Introduction. Material and methods should present methodology used specifically for this study and in much greater detail than it is now.*

**Response:** A summary of the main features of the models was added in the form of a table in Material and methods, while a short paragraph about the models was added to the Introduction. Also, the full description of the models has been modified and transfered to the Supplement S2. The main focus has been placed on the methodology which is now described in greater detail.

*Line 137. Please give a reference to the ALADIN/HR atmospheric model.*

**Response:** The references Tudor et al. (2013, 2015) were added.

*Lines 143-144. 'In this model, the horizontal ROMS grid resolution is 2 km and there are 20 vertically spaced sigma levels controlled by the following parameters ...'. Ocean model results are strongly dependent on the vertical resolution of a model. This is hinted by the authors in Lines 622-623 ' In addition, the lack of vertical resolution in the ROMS-full model probably contributes to the improper representation of the dense water dynamics'. Has the sensitivity study been performed by the authors or other researchers? The authors attempt to compare 'the types of the models' however the skill of the same model can change significantly when model governing parameters or boundary conditions (e.g. river discharge) are changed. The authors should demonstrate that parameters of the models used for comparison provide the best results within the limitation of the specific 'type' of model in order to suggest which 'type' of the model is the best.*

**Response:** A sensitivity study has not been performed as we only use already well validated models – i.e., the skills of each model have previously been assessed (= the best parametrization possible are used) and this is not the aim of this study to redo this assessment. However, the forcing used in the different models (rivers, boundary conditions, etc.) are discussed at length in the paper.

*Lines 201-206. Have you noticed the 'double penalty effect' ( see e.g. https://doi.org/10.5194/os-16-831-2020 ) in the higher resolution models? If yes, how it impacts on the final results?*

**Response:** A comment has been added to subsection 3.1. as follows: „Lastly, the comparison of the performance of models with different resolutions may be affected by the double-penalty effect (Crocker et al., 2020) meaning that in pointwise comparison with observations the finer resolution models tend to be penalised more than the models with coarser resolution and therefore they can verify worse. When a model has sufficient resolution to reproduce a small-scale feature but it simulates it incorrectly, it is penalised twice: once for not simulating the feature where it should have been and once for simulating it where it hasn't been observed. Contrarily, if a model resolution is not sufficient to reproduce a feature, it will be penalised only once for not reproducing the feature. This might partly explain why the AdriSC-ROMS model presents a higher bias variability in both temperature and salinity and thus lower standardized deviations than the ROMS-hind and ROMS-full models."

*Line 202-203. 'In order to compare different simulations, model results with grid resolution coarser than 1 km are interpolated to the AdriSCROMS 1 km grid…' .Does it mean that all model outputs were also interpolated in the vertical to the AdriSCROMS sigma-coordinate grid with 35 levels? Please clarify.*

**Response:** Model outputs were not interpolated in the vertical, thus the sentence was modified to: „In order to compare different simulations, model results with horizontal grid resolution coarser than 1 km are interpolated to the AdriSC-ROMS 1 km grid...".

*Line 204-206' For the ocean simulations, MEDSEA, ROMS-hind and ROMS-full results are regridded to 1 km resolution, while for the atmosphere, ERA5, ALADIN/HR-hind, ALADIN/HR-full and AdriSC-WRF results are all regridded to 1 km resolution.' The use of the word 'while' is strange as all model outputs were regridded to the same scale. Please re-word.*

**Response:** The sentence has been modified to: "More specifically, the results of the ocean models (MEDSEA, ROMS-hind and ROMS-full) and atmospheric models (ERA5, ALADIN/HR-hind, ALADIN/HR-full and AdriSC-WRF) are all regridded to a horizontal resolution of 1 km".

*Line 214-215. ' probability density functions of the biases (i.e., differences) between the results of the simulations and the in-situ temperature and salinity observations' . This is the core component of methodology and it has to be described in much more detail. How the bias is calculated? If it is the average of all differences at all locations and all times then it would be impossible to calculate the PDF. Or is it an average of daily differences? Or something else?*

**Response:** An explanation of the bias calculation has been added to the Methods section as follows: „The biases are calculated as differences between the daily results of the simulations and the available observations (i.e., they are daily instantaneous bias errors). Consequently, the model results are extracted at the location (i.e., near neighbour grid point), depth (i.e., linear interpolation from model depths to observation depth) and timing (i.e., approximated to daily average) of the observations. The biases are then obtained as the difference between model results and observations at each point in time, depth and space."

*Line 218. 'The probability density functions are obtained with a kernel-smoothing method..' Which kernel was used? What was the size of the smoothing window? The results may be sensitive to these factors. Please give more details here (including a reference) as otherwise your results cannot be replicated. An estimate of the sensitivity of results to the size of the smoothing window will be helpful.*

**Response:** The description of PDFs has been expanded as follows: „The probability density functions are obtained with a kernel-smoothing method (Bowman and Azzalini, 1997) which calculates the probability density estimate based on a normal kernel function, and is evaluated at 100 equally-spaced points".

*Line 223. 'minimum turbulent heat fluxes in the atmosphere'. Do you mean 'minimum downward turbulent heat fluxes in the atmosphere at the sea surface' Please clarify.*

**Response:** Yes, minimum downward turbulent heat fluxes in the atmosphere at the sea surface are presented. Therefore, the sentence was modified to: „... the minimum downward turbulent heat fluxes in the atmosphere (at the sea surface),...".

*Line 230. 'bottom PDA time series is presented without the seasonal signal which is removed from the series using the least-squares method.' Please give more details of how PDAs are calculated otherwise the statement in Line 232 'the time evolution of the spatial distributions of the bottom PDAs' is difficult to comprehend.*

**Response:** The PDA calculation has been described in greater detail in the Methods section as: „In addition, the daily bottom PDA time series are presented without the mean and the seasonal signal (yearly and half-yearly) which are removed from the series at each point. More specifically, after subtracting the mean and detrending the time series, the seasonal signal is calculated using the least-squares method of a sine function and subtracted from the series. The final time series without the seasonality is obtained by adding the trend".

*Lines 233-234. 'An additional analysis (only presented and commented in Supplementary Material) quantifies the total daily volume transport of the outflowing dense waters..' The near-bottom transport of dense waters is a major parameter quantifying the intensity of dense water cascades. Therefore it has to be included in the main text ( both results and discussion) in sufficient detail.*

**Response:** The section about dense water transport has been included in the main text. Also, the methodology has been expanded.

*Lines 238-248. The validation of the four models against CTD casts is very helpful. In order to help a reader to interpret the figures given in this section, the methods of calculating biases given in the previous section have to be presented in much more detail. It is advisable to extend the basic stats ( mean and standard deviation) to include more advanced tools of model validation, e.g. Pearson correlation, Willmott skill parameter or Taylor diagram.*

**Response:** The validation statistics has been extended with a Taylor diagram (Figure 2a) and a paragraph with the description has been added to the subsection 3.1.

*Lines 238-255. The four models have different resolutions and some of them may not resolve the processes of the scale of the baroclinic Rossby radius. Please provide a map of Rossby radius for your area. You may wish to use a simplified method presented in Chelton et al . 1998: Geographical variability of the first-baroclinic Rossby radius of deformation. J. Phys. Oceanogr., 28, 433-460.*

**Response:** A map and time series of the Rossby radii have been added to the article as Figure 14 and described in subsection 4.2.

*Lines 301-302. 'Overall, for all models, maximums of wind stresses are associated with bora events, while downward turbulent heat fluxes seem to be more influenced by the seasonal variations of the sea surface temperature' . Please clarify the second part of this statement. From the qualitative point of view, stronger and colder winds (the bora) should have a greater influence on downward heat fluxes from the atmosphere to the ocean than smooth and therefore weaker seasonal variations.*

**Response:** The following explanation has been added: „Overall, for all models, maxima of wind stresses are associated with bora events, while downward turbulent heat fluxes seem to be influenced by the seasonal variations of the sea surface temperature (SST) more than the wind stresses. In other words, the largest input to the downward turbulent heat fluxes is coming from

the bora wind, yet a small fraction – which is found to influence maxima of the heat fluxes – is coming from SST. That is the reason why maxima of heat fluxes occur mostly during bora episodes in late December/early February (Fig. 4), whereas the maxima of wind stresses occur mostly during bora episodes in early February/early March (Fig. 3)."

*Lines 345-416. The presentation of results is mostly concentrated on the atmosphere-ocean heat fluxes and the processes of formation of dense water. However, in contrast to CTD observations the models provide an opportunity to calculate dense water transport, which is a key component of cascading. This sections gives a good quantitative description of heat fluxes, while it describes the transports only qualitatively. This omission has to be rectified.*

**Response:** The Authors agree and a quantitative description of transports has been added to the main article.

Dear Referee #2,

Thank you very much for your review and comments which have been addressed in the sections below.

*The last line of the abstract is a bit strong in the sense that the article does not prove that 1-the spatial scale, 2-non hydrostatic, 3-fine resolution and 4-forcing at the open boundary are pre-requisites, it only shows us that one simulation, of the four taken into account, is better than the others and it is closest to the dynamic theory of the behavior of the northern Adriatic basin.*

**Response:** The abstract is modified as follows: „The prerequisites for proper modelling of the ocean circulation in the Adriatic basin, including kilometre-scale atmosphere-ocean approach, non-hydrostatic atmospheric models, fine vertical resolutions in both atmosphere and ocean and the location and forcing of the open boundary conditions, are thus discussed in the context of the analysed simulations".

*The calculations of the probability density functions should be given in more detail, perhaps a sensitivity study of the method should be mentioned.*

**Response:** The description of the calculation of PDFs has been expanded as follows: „The probability density functions are obtained with a kernel-smoothing method (Bowman and Azzalini, 1997) which calculates the probability density estimate based on a normal kernel function, and is evaluated at 100 equally-spaced points".

*Also, the seasonal signal removed from the PDA time series would be shown.*

**Response:** As the models are only compared for 1 year simulation, the seasonal cycle is approximated by a least-squares method of a sine function (as usually done when dealing with sparse measurements). Consequently, the authors believe that there is no much sense of presenting the seasonal signal. However, to clarify the methodology used, the following description has been added to the method section: „In addition, the daily bottom PDA time series are presented without the mean and the seasonal signal (yearly and half-yearly) which are removed from the series at each point. More specifically, after subtracting the mean and detrending the time series, the seasonal signal is calculated using the least-squares method of a sine function and subtracted from the series. The final time series without the seasonality are obtained by adding the trend."

*The numerical models used in the study have different resolutions, so a comment regarding the numerical resolution of processes with different Rossby's radius deformation would be important.*

**Response:** A paragraph about the resolution and the Rossby radii has been added to subsection 4.2.

*Vertical resolution can also play a large role in improperly representing the dynamics of dense water. Therefore, the same model can change significantly as the parameters that govern it vary as well as the boundary conditions. This should be described in more detail even if not verified.*

**Response:** The forcing used in the models (rivers, boundary conditions, etc.) are discussed at length in the paper. The influence of the vertical resolution is mentioned in section 4.2.

*Also, the 31-years simulation should be described in more detail.*

**Response:** The description of the 31-year simulation is expanded in Introduction.

*The authors are very focused on heat fluxes, but a good model would also allow to estimate the transport of dense water*

**Response:** More focus hase been placed on the transport of dense water. The quantitative assessment of the transport has been added to the main article as subsection 3.3.3.

---

## Author Response (AR2)

Dear Editor,

thank you very much for your detailed and helpful comments which have been addressed in the sections below.

*Too frequently you use of "low", "high", "above", below" etc. for quantities unrelated to elevation. Some specific cases are noted below but there are many other cases in the text. Related to this, the negative signs for heat flux values makes difficulties in comparing them as large or small. Fluxes from sea to atmosphere would be positive and comparisons less confusing.*

**Response:** The stated adjectives have been replaced by appropriate adjectives for quantites unrelated to elevation. Also, the sign of the turbulent heat fluxes has been changed to positive (i.e., to upward) and figures (4 & 6) and corresponding text have been modified.

*Line 13. Last "the" –> "a" and line 14, "the" –> "a"*

**Response:** „the" has been replaced by „a" in both lines.

*Line 18. "The prerequisites" –> "Likely prerequisites"? [The reviewers might still complain that you have not proved that these aspects are necessary as you assume, although most would agree that they are.]*

**Response:** „Likely" was added to the sentence.

*Line 21. "the 31-year long evaluation run of the Adriatic Sea climate model" –> "a 31-year long run of the fine-resolution Adriatic Sea climate model"? (to avoid "evaluation run", undefined at this stage).*

**Response:** The sentence has been changed accordingly.

*Line 33. "the water masses" –> "water"?.*

**Response:** „the" was deleted.

*Line 37. Better "NAdDW is known . ."*

**Response:** The sentence is modified accordingly.

*Line 40. ". . mountain range (e.g.) . ."?*

**Response:** „range" was added to the sentence.

*Line 87. "the 31-year" –> "a 31-year" (because not yet defined in this Introduction).*

**Response:** „the" was replaced with „a".

*Line 90. Better "several-decade-long".*

**Response:** „several decade long" has been changed to „several-decade-long".

*Line 92. "contrarily to" –> "contrary to" or "as distinct from"*

**Response:** „contrarily to" has been changed to „contrary to".

*Table 1. You could reduce the line spacing and perhaps the column width within the Table so that it fits better.*

**Response:** The line spacing and column width in Table 1 have been reduced.

*Line 140. To satisfy Referee 1 you might add explicitly "Model outputs were not interpolated in the vertical."*

**Response:** The suggested sentence was added at the end of the paragraph.

*Line 145. Better ". . in this study "dense waters" are defined as having PDAs equal or larger"?*

**Response:** The sentence has been modified accordingly.

*Line 153. Better "Consequently" –> "I.e." or "That is" or "Specifically" or "More precisely"?.*

**Response:** „Consequently" has been replaced by „That is".

*Line 206. "an overestimation of the temperatures in MEDSEA" seems to contradict the MEDSEA median bias 0.00 (line 203).*

**Response:** In order to clarify, the snetence has been modified as follows: „Median temperature bias is smaller in AdriSC-ROMS and MEDSEA, but they have the largest MAD due to an overestimation of temperatures in MEDSEA (up to 4.5 °C) and both an over- and underestimation of the temperatures between -2 and +2 °C in AdriSC-ROMS".

*Line 221. "a higher" –> "larger" and "lower" –> "smaller". However, larger variability leading to smaller standardised deviations is a puzzle. How are the deviations "standardised / normalised"?*

**Response:** „higher" has been replaced by „larger". The last sentence has been modified to: „This might partly explain why the AdriSC-ROMS model presents a larger bias variability in both temperature and salinity than the ROMS-hind and ROMS-full models". The normalized standard deviations were calculated as the ratio of the standard deviation of model results and standard deviation of observations. The explanation was added to the Methods section. Also, the diagram (Fig. 2) and the text have been corrected because square root wasn't taken in the original calculation (thus variance was plotted originally, not standard deviation).

*Line 242. "highly" –> "strongly"*

**Response:** „highly" has been replaced by „strongly".

*Line 243. "return" –> "turn"*

**Response:** „return" has been changed to „turn".

*Line 250. "above" –> "less in magnitude than"?*

**Response:** „above" has been replaced by „less than".

*Lines 255, 262. "below" –> "larger in magnitude than"?*

**Response:** „bellow" has been replaced by „larger than".

*Line 263. "lower" –> "smaller"*

**Response:** „lower" has been replaced by „smaller".

*Line 264. "below" –> "less than"*

**Response:** „below" has been changed to „less than".

*Line 267. "above" –> "less in magnitude than"?*

**Response:** „above" has been replaced by „less than".

*Line 274. "highest" –> "strongest". "lowest" –> "smallest" or "weakest"*

**Response:** „highest" has been changed to „strongest" and „lowest" has been changed to „weakest".

*Line 275. ". . smallest heat loss."*

**Response:** The sentence is modified accordingly.

*Line 349. "Consequently" –> "That is"?*

**Response:** „Consequently" has been changed to „That is".

*Line 355. "6b" –> "7b"*

**Response:** The mistake was corrected.

*Line 396. Unclear what is "also (shown?) by ROMS-full"*

**Response:** The structure of the sentence has been corrected as follows: „. However, in the Jabuka Pit, MEDSEA shows PDA values up to 29.5 kg m$^{-3}$ in November 2014, slowly decreasing to 29.2 kg m$^{-3}$ before the first bora event, whereas ROMS-full produces values below 29.25 kg m$^{-3}$ around 20 December 2014".

*Line 403. "of the" –> "in"?*

**Response:** „of the Kvarner Bay" was corrected to „off-shore from the Kvarner Bay".

*Lines 462, 464. Omit "down to"?*

**Response:** „down to" was omitted.

*Lines 466, 468, 473. Better "highest" –> "largest". Similarly lines 478, 479 ("low" –> "small").*

**Response:** „highest" has been replaced by „largest" and „low" by „small".

*Line 538. "highest" –> "largest". Line 539; "lower" –> "less". Etc.*

**Response:** „highest" has been replaced by „largest" and „lower" by „less".

*Line 544. ". . varies between 0.5 and 1.0 km . ." or simply ". . is 0.5-1.0 km . ."*

**Response:** The sentence has been modified accordingly.

*Line 546. ". . very small (less than 500 m) . ."*

**Response:** The sentence has been modified accordingly.

*Lines 547-548. ". . 4 km to extremely small values . ."*

**Response:** The sentence has been modified accordingly.

*Line 553. "below" –> "finer than"*

**Response:** „below" has been replaced by „finer than".

*Line 562. ". . (by 20-40 m). . ."*

**Response:** „by" was added.

*Line 611. "contrarily" –> "contrary"*

**Response:** „contrarily" was changed to „contrary".

*Lines 647-648. Better ". . (1) kilometre- or finer-resolution atmosphere-ocean models and non-hydrostatic atmospheric models, . ."*

**Response:** The sentence has been modified accordingly.

*Line 679. "Middle-range" –> "Medum-Range". Line 680 "Forecasts . ."*

**Response:** The name has been corrected.

*Figure 6 caption. Better ". . series of wind stresses and daily turbulent fluxes averaged . ." (order as in figure)*

**Response:** The caption has been modified accordingly.

*Supplement*

*S2.1, paragraph "Nucleus . ." line 5. "Medium-Range Weather Forecasts . ."*

**Response:** The name is corrected accordingly.

*S3.1, line 3 of text: "largest" –> "highest" – conventionally "low" and "high" are used with temperature!*

**Response:** „Low" and „high" convention is applied for temperature.

*S3.2, page 9 line 5. "form" –> "than in"*

**Response:** „form" has been replaced by „than in".

---

## Author Response (AR3)

Dear Editor and Referees,

thank you for your final comments and corrections which have been addressed in the sections below.

*REFEREE 1. "My only minor comment is that the Conclusion section is too lengthy to my liking. Some of the text can be transferred to the Discussion section. The authors may follow the examples below from JGR-Oceans and JPO:*
*DOI: 10.1029/2021JC018161*
*DOI: 10.1175/JPO-D-19-0134.1*
*DOI: 10.1029/2022JC019055"*
*My (EDITOR's) comment: I think all the present content of the "Conclusions" section is valid, but perhaps the content of the bullet points is quite detailed. I don't think the bullet points are exactly "Discussion" but some of their content might appear usefully as a summary at the end of the "Results" section; content depending on the "Discussion" should remain in "Conclusions". The "Conclusions" could then point to that summary. I leave this to you (the authors) to decide what to do (if anything).*

**Response:** A slight modification has been made in which a part from the „Conclusions" has been moved at the end of the „Results" section as follows: „However, in AdriSC-ROMS extreme dense waters are generated homogeneously over the entire northern Adriatic, while they appear as patches in ROMS-full in which a maximum is found off the southern tip of Istria, along the Senj Jet", while the "Conclusions" section is now as follows: "Also, extreme dense waters are generated continuously in time and over the entire northern Adriatic in AdriSC-ROMS, while they appear as patches in ROMS-full".

*REFEREE 2. "I really appreciated the considerable editing and rewriting of the article.*
*My overall comment is very good and the paper can be published.*
*I would like the authors to meditate on the role of the evaporation fluxes (E-P) in the process of producing dense water which instead seems to be mainly conducted by 'only' cooling.*
*A small discussion, with a couple of sentences, on the role of numerical conservation of the total volume of the Adriatic respect to the volume of the new and dense water produced, taking into account the different role of evaporation (E-P) compared to cooling, could be a good link also for future articles."*
*My (EDITOR's) comments: It should be clarified whether latent heat flux due to E-P is included in the cooling rates stated. Indeed E-P affects total volume but only a couple of sentences are suggested.*

**Response:** A small discussion has been added to Sect. 4.3 as follows: „Finally, besides the river discharges, the surface freshwater E-P fluxes (i.e., Evaporation – Precipitation fluxes) also determine the surface salinity of the northern Adriatic Sea. The E-P fluxes are taken into account in all ocean models presented in this study but are derived from really different atmospheric models. Consequently, the difference in dense water results can also be influenced by the differences in E-P fluxes."

*EDITOR's "Technical corrections".*
*Line 193. Please check the depth ranges; there seems to be a gap between 100 and 200 m.*

**Response:** Yes, there was a gap in the 100-200 m range, thus the sentence has been corrected to: „The number of observations depending on the depth is: (1) 7698 for the 0-50 m range, (2) 7582 for the 50-200 m range, (3) 2130 for the 200-500 m range, and (4) 1577 for the 500-1200 m range".

*Line 201. Better "lower" –> "poorer".*

**Response:** „lower" has been replaced by „poorer".

*Line 258. Better "high" –> "strong".*

**Response:** „high has been replaced by „strong".

*Line 263. "-1100" –> "1100"?*

**Response:** „-1100" has been corrected to „1100".

*Line 544. "3c" –> "14c"*

**Response:** „3c" has been corrected to „14c".